# Landscape Agroecology. The Dysfunctionalities of Industrial Agriculture and the Loss of the Circular Bioeconomy in the Barcelona Region, 1956–2009

**Claudio Cattaneo [1,2,*], Joan Marull [2] and Enric Tello [3]**

[1] Department of Environmental Studies, Faculty of Social Studies, Masaryk University, Jostova 10, 602 00 Brno, Czech Republic

[2] Barcelona Institute of Metropolitan and Regional Studies, Autonomous University of Barcelona, 08195 Bellaterra, Spain; joan.marull@uab.cat

[3] Department of Economic History, Institutions, Policy and World Economy, University of Barcelona, 08034 Barcelona, Spain; tello@ub.edu

\* Correspondence: claudio.cattaneo@uab.cat; Tel.: +34-93-586-8862

**Abstract:** The paper analyses how between 1956 and 2009 the agrarian metabolism of the Barcelona Metropolitan Region (BMR) has become less functional, losing circularity in biomass flows and in relationship to its landscape. We do so by adopting a Multi-Energy Return on Investment (EROI) and flow-fund (MuSIASEM) analyses and the nexus with landscape functional structure. The study of agricultural flows of Final Produce, Biomass Reused and External Inputs is integrated with that of land use, livestock, power capacity, and population changes between 1956 (at the beginning of agrarian industrialization) and 2009 (fully industrialized agriculture). A multi-scale analysis is conducted at the landscape scale (seven counties within the Barcelona metropolitan region) as well as for the functions deployed, within an agroecosystem, by the mutual interactions between its funds (landscape, land-uses, livestock, and farming population). A complex nexus between land, livestock, dietary patterns, and energy needs is shown; we conclude that, from the perspective of the circular bioeconomy the agrarian sector has gone worse hand in hand with the landscape functional structure. Therefore, a novel perspective in landscape agroecology is opened.

**Keywords:** landscape agroecology; MuSIASEM; Multi-EROI; circular bioeconomy; Barcelona Metropolitan Region; industrial agriculture

## 1. Introduction

Agrarian industrialization has allowed for unprecedented improvements in land and labour productivity, but, in such a production-oriented perspective, many costs have been overseen [1,2]. Since the production process, like in an industrial system, is conceived as a linear and highly specialized one—i.e., by increasing inputs to increase output- the agro-ecological practices [3] of traditional organic agriculture have been left behind. These were centred on the multi-functionality of biomass flows and on an equilibrated interdependency—i.e., by means of a mixed farming—of its elements (i.e., cropland, pastureland, forestland, livestock, power capacity, and farmers).

This paper opens a new perspective in landscape agroecology [4,5], one that envisions agroecological landscapes [6]. From a sustainability perspective, this is an important issue because the combined effect of agro-industrialization and dietary change—namely, more meat for a cheaper price—has considerable hidden costs in terms of energy efficiency, landscape ecology, bio-cultural heritage, biodiversity, climate change, soil and water quality, and human nutrition and health [6].

The purpose of this work is to demonstrate, when dealing with issues of sustainability and agriculture, the relevance of the landscape agroecology approach (as an important fund for farm systems): this is an integrated approach that conceives sustainable agriculture not simply as organic or local agriculture, but as a set of meaningful practices at the landscape scale, which is where all types of ecosystem functions and services play a role. For instance, little has to be said about the sustainability of organic agriculture if this depends on inputs that are not interwoven with the local agroecosystem, or if it provokes land use changes that destroy habitats and bio-cultural heritages, or if harvest by-products have lost any value [7]. In other words, sustainability in agriculture cannot be conceived without the recognition of the complex relationships stemming from agroecological practices imprinted in the landscape.

The general hypothesis is that the land-cost of sustainability [1], which is an environmentally extended example of the social costs of business enterprise [8], is not being recognized in current policies and market dynamics. Our specific hypothesis is that sustainability can be assessed by the use of flow-fund models and the related energy and landscape efficiency indicators [9–11]. In particular, when the modelled agro-industrial practices and land uses break certain flow-fund relationships between the agricultural flows and the funds from which they originate, multiple losses in ecosystem services, in energy and in landscape efficiency, are observed.

The current state of research in landscape agroecology is set in the outcomes of the Sustainable Farm Systems research project, which has focused on an innovative development of Energy Return on Investment (EROI) analysis [6,11–13] also merging it with MuSIASEM [14], on nutrient cycles [15,16] and on the original development of Energy–Landscape Integrated Analysis [10,17,18], which in turn draw from previous work on social metabolism [9,19,20] and EROI analysis [21]. As well, these novel methods applied in landscape agroecology can bridge with and contribute to the land sharing/sparing debate [22], which is still a controversial issue [23,24].

In particular, we learn from Tello et al. [11] that biomass flows within agro-ecosystem have to be balanced between incoming-outgoing and recirculating ones, and that industrial agriculture has favoured maximizing linear flows (input-output) over the ones that recirculate internally. However, little is said about the relationships between these flows and the different funds within an agroecosystem. In this paper, we fill this research gap by applying the flow-fund analysis of MuSIASEM and show that the balance between the different flows that enter into, recirculate across and exit from the different funds (i.e., livestock and land-uses) is also important and it should be studied at the landscape scale (a unit of analysis that is necessary to close the metabolic cycles of the agroecosystem).

The main aim of this paper is to assess, for the case study of the Barcelona Metropolitan Region, the loss in sustainability that has occurred as a result of agricultural industrialization by highlighting the virtues of a past model that could inspire future developments towards more sustainable agriculture, landscape, and diets. In conclusion, we propose multi and inter-functionality of farm practices within their landscape, as well as the need for dietary change away from industrial meat production.

The paper is structured in the following way: next section presents the case study, which is the BMR, composed of seven counties and 164 municipalities for the years 1956 and 2009, the materials and sources used for the analysis and the methods employed. Section 3 presents the results that integrate the Multi-EROI analysis [11,12,25,26] and the MuSIASEM analysis [9], with a specific focus on the nexus between the main agroecosystem funds, the intensity of their flows and the variation between 1956 and 2009. Section 4 discusses the results, in particular, how the proportions between the main funds have changed, allowing for an increase in the relative productivity of specific products—meat in particular—but at the cost of a disintegration of the flows connecting these funds, so that a shift has occurred from a circular flowing of matter-energy towards a "linear" one, resembling an input-output industrial system of a lower agro-ecological quality. It finally presents the results in the light of landscape agroecology. Section 5 concludes.

## 2. Materials and Methods

### 2.1. The Case Study

The Barcelona Metropolitan Region (BMR) is a very densely populated area and is the sixth largest urban area in Europe [27,28]. It is composed of seven counties. Two at the centre, on the coast: Barcelonés—the smallest and most populated—and, to the west, Baix Llobregat with an important agrarian park, together they make most of the Barcelona Metropolitan Area (BMA), which is the most urbanized part of the BMR. At the centre, off the coast is Vallès Occidental, with also a large population; to the north-east are Maresme, on the coast and Vallès Oriental inland; and, to the south-west, Garraf, on the coast and Alt Penedés inland (Figure 1).

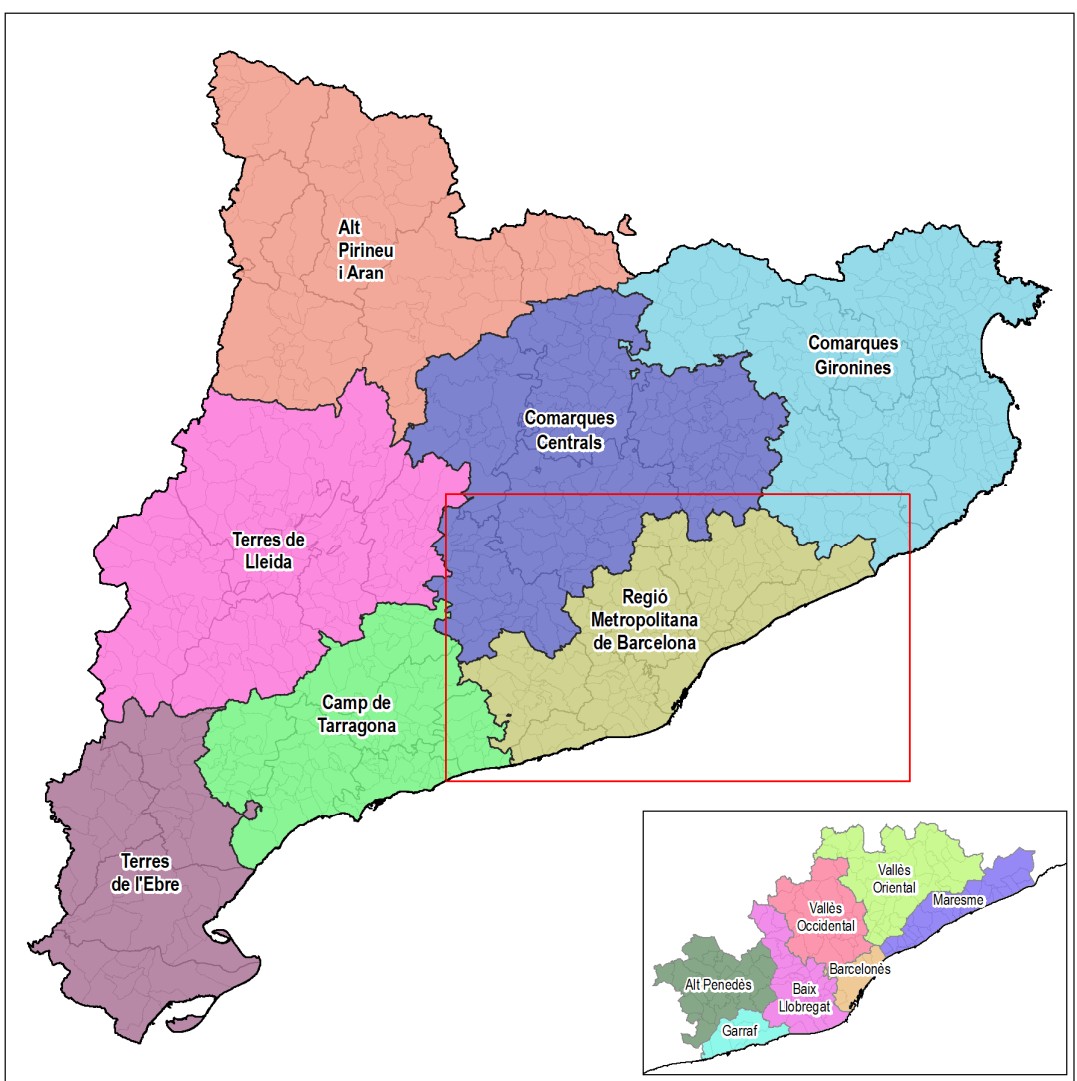

**Figure 1.** The seven counties of the Regió Metropolitana de Barcelona (BMR) within the seven regions of Catalonia. Source: Authors' own elaboration.

Population in the BMR has nearly doubled, reaching five million, more than 1500 inhab./km². Urban area has grown even more, mainly at the cost of agricultural land. Recently, there has been a growing public interest towards the implementation of urban and peri-urban agriculture [29], the need for policies oriented to the re-ruralization of the city, the foment of an urban food policy [30], and new territorial planning that considers restoring cropland as much as possible with respect to 1956 levels [31].

*2.2. Methods*

We use the flow-fund approach, as proposed by Georgescu-Roegen [32] and developed, adopting the concept of social metabolism, by [9,33]. According to Gerber and Scheidel [34], analysis of social metabolism is fundamental towards substantive economics and the approach by [10,11]— which we undertake in this paper—represents an interdisciplinary development that combines it with other methods such as landscape ecology.

We analyse the social metabolism of agroecosystems. Here, we consider landscape, farmland, livestock, farmers, and machinery as funds, which are capable to provide flows (food, feed, fibre, fuel, finance, as well as unharvested biomass for associated biodiversity). The underlying assumption is that funds are capable of providing flows only at a given rate (i.e., a forest can provide flows of timber equivalent to its growth rate). If the flows are larger, then funds are overexploited in a non-sustainable way (i.e., deforestation). Also, flows, in order to be optimized, require maintenance and care of the correspondent funds (i.e., a cow cannot give milk if not fed, or a unit of land cannot produce if soil fertility and biota are not maintained, or a tractor cannot function properly if not serviced regularly). Funds that are living systems require biomass for their maintenance and reproduction; inorganic funds—such as machinery—require non-renewable materials for their construction and maintenance (The system boundaries of our energy analysis are the following: we take a farm-gate approach [11]. This means that we account for all the energy flows that occur within the agroecosystem considering a farmer's standpoint [11]. This means that when a product exits the farm (i.e., a pig to be slaughtered), we account for the energy content of the whole produce (i.e., the entire animal weight). When a biomass product is imported to the farm, we account for the embodied energy in the transportation from the previous agroecosystem's gate (i.e., the energy embodied in the transport of animal feed from the original agroecosystems they have been produced). On the other hand, when a non-biomass product is imported to the farm as an artificial input (be it a fund, like a tractor, or a flow, like biocides), we account for their energy content plus the embodied energy in their production and delivery, based on life cycle accounting, as detailed in [35]).

Specifically, we compare the agroecosystems of the BMR for the years 1956 and 2009 across different scales. The funds analyzed are land, in particular, woodland, pastureland, cropland (green crops, vineyard, and other woody crops), shrub land, and built-up land (urban and transport infrastructure); livestock (equids, bovines, sheep and goats, swines, poultry, and rabbits); population (total inhabitants and farmers); and, machinery power capacity. The flows we consider are divided between Final Produce (from forestry, from cropland and from livestock, i.e., timber, firewood, vegetable, and animal products) and Biomass Reused (that maintains funds, i.e., seeds, reploughed biomass, manure, feed, litter) and Unharvested Biomass (herbivory, weeds, and under-exploitation of certain funds, such as present day forests).

Table 1 below lists the sources, of funds, and flows analysed, at different geographical scales, for 1956 and 2009; flows not specified in the table have been modelled; information that is only available at scales larger than the municipal has been converted to the municipal level following a weighting process; please refer to the Supplementary Information for the details and specific process.

**Table 1.** Sources of funds and flows data.

| Data Sources Year 1956 (*Flows Are in Italics*) | Municipal | County | Provincial | National |
|---|:---:|:---:|:---:|:---:|
| Land use from GIS | Υ | | | |
| Land use (ha) from yearbook | | | Υ | |
| *Land flows (harvest in weight)* | | | Υ | |
| Animal census | Υ | | | |
| *Animal products (meat, milk, egg, wool)* | | | Υ | |
| Total Population | Υ | | | |
| Farmers population | | | Υ | |
| Machinery | Υ | | | |
| *Fertilizers biocides* | | | | Υ |

| Data Sources Year 2009 (*Flows Are in Italics*) | Municipal | County | Provincial | National |
|---|:---:|:---:|:---:|:---:|
| Land use from GIS | Υ | | | |
| Land use (ha) from yearbook | | Υ | Υ | |
| *Land flows* (*harvest in weight*) | | Υ (forest products) | Υ | |
| Animal census | Υ | | | |
| *Animal products* (*meat, milk, egg, wool*) | | | Υ | |
| Total Population | Υ | | | |
| Farmers population | Υ | | | |
| Machinery | Υ | | | |
| *Fertilizers biocides* | | | Υ | |

The analysis is then carried out calculating a set of different EROI [2,11,26] and other flow-fund and fund/fund indicators based on the MuSIASEM approach [9]. In particular, the EROI presented are (i) the Final EROI (FEROI), relating the Final Produce (FP) and the Total Inputs Consumed (TIC), which are the sum of Biomass Reused (BR) and External Inputs (EI); (ii) the External Final EROI (EFEROI), relating FP/EI—which represents the relationship between the output and the input to the farm; (iii) the Internal Final EROI (IFEROI), relating FP/BR, which indicates the biomass recycling effort. Then, we analyze (iv) the NPPEROI, which relates Net Primary Productivity with the sum of total inputs consumed (TIC) and unharvested biomass; (v) the Agroecological EROI, which relates FP to the sum of unharvested biomass and TIC; (vi) Biodiversity EROI, which relates unharvested biomass to the sum of TIC and unharvested biomass. Finally, we consider (vii) Land Final EROI and (viii) Livestock Final EROI representing the relationship between the Final Produce of, respectively, the Land and Livestock subsystems with the correspondent internal and external inputs. All of these indicators relate different flows with each other, but in social metabolism, it is important to relate these flows to their correspondent funds (i.e., the flow of final produce per hectare, or per farmer, or per unit of livestock), as well as the relations within or across funds (i.e., the size of cropland with respect to total land, or the density of livestock per unit of cropland). By all of these flows and funds relationships, we are exploring the nexus between farmland and livestock functions. Figure 2 shows the nature of energy flows inside and outside the agroecosystem (EI and FP), and between its compartments (BR from cropland, pastures, and forests to livestock and BR from livestock back to cropland). Figure 2 shows how energy and biomass flows go through and across different elements of an agro-ecosystem. Of particular interest are flows of reused biomass, from farmland (cropland, pastures, and forests) to livestock and of livestock services back to cropland. As well, external inputs and final products enter and exit the agroecosystem's farmland and livestock compartments.

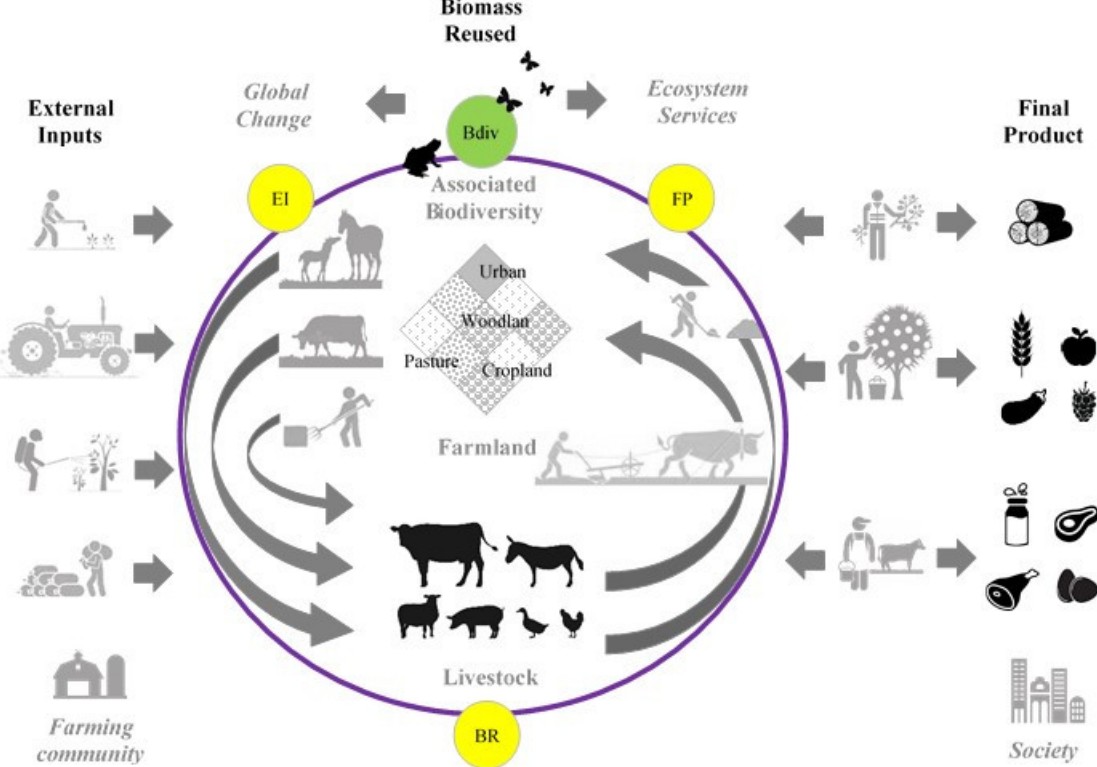

**Figure 2.** representation of an agro ecosystem's multi-functionality. Source: Authors' own elaboration.

Landscape Agroecology Metrics

We use two land metrics to evaluate landscape patterns and processes. The Shannon index (H) that is applied to the land-cover structure is used as an indicator of landscape's heterogeneity, and as a proxy for habitat differentiation that hosts biodiversity [36]. The higher its values, the more equally distributed are land covers. Then, from a landscape agroecology perspective, we adopt a variation of it—the geographical distribution index—to create a proxy also for agricultural multi-functionality. The formula, which is common to both the original and its variation, is:

$$H = -\sum p_i \log_k p_i \tag{1}$$

in which, if we refer to the distribution of land covers in a territorial unit of analysis, k are the different land covers of a territorial unit of analysis (i.e., the Alt Penedès county) and p is the size of land cover i expressed in relation to the unit of analysis. In its landscape agroecology variation, we apply the same formula to show how a fund (i.e., woodland land cover, or poultry animal typology, or population) is distributed across k regions, where p is the size of the fund in region i with respect to the total sum of that fund over the k regions.

The index ranges between 0, representing maximum landscape homogeneity, which is one land cover in the unit of analysis, or, in its variation, the fund entirely concentrated in one region; and 1, representing maximum landscape heterogeneity through the equal distribution of land covers in the unit of analysis or, in its variation, equal distribution of the fund in each region of the BMR.

The second indicator we adopt is the Ecological Connectivity Index (ECI) [17], related to landscape functionality, which measures in a [0,1] range the capacity for connecting flows of biomass and information across a territorial unit of analysis, which is fundamental for supporting biodiversity and related ecosystem services: the higher the index, the more connected is the landscape, so that biodiversity can move more freely.

## 3. Results

For simplicity of presentation of results, the analysis is done at the county and BMR level. First, the evolution of the main funds is presented, then the EROIs indicators and finally their flow-fund representations. From the perspective of sustainability indicators, we look at social and environmental sustainability, assuming that the present industrial system of agriculture is focusing only on financial viability. The result is both an environmental problem (energy inefficiency and loss of habitats), as well as a social one (loss of cultural landscapes). The EROIs, the variations in flow-fund and in across funds relationships, and the landscape metrics indicate these sustainability losses.

*3.1. Cropland Loss, Livestock Growth, Mechanization and Urbanization*

Across the BMR cropland area has gone down from 40% to 18% of the territory, losing it to urban area, which is up from 5% to 23%, and to woodland that has grown from 37% to 42%. Table 2 shows the evolution in the area of urban and farmland categories in each county and the landscape agroecology indicators. The Shannon index accounts for six land covers: green crops, wood crops, vineyards, meadows, shrub land, and woodland; between 1956 and 2009, their landscape heterogeneity has gone down in all counties. The variation in the Ecological Connectivity Index shows that it has always gone down too. Finally, the geographical distribution of each land use is presented in the bottom line. The index, which applies the Shannon Index formula for 1956 and 2009, shows how the surface of each land use is distributed across counties and indicates that in 2009 meadows and cropland—and all of its sub-categories—were less evenly distributed—the Index falls from 0.96 to 0.90 (meadows) and from 0.91 to 0.78 (cropland). In particular, the geographical localization of vineyards is increasingly skewed towards Alt Penedès: this is the crop category, which has decreased the least in surface, but it has reached the most uneven distribution. On the other hand, woodland and urban areas are more evenly distributed in the BMR than they were in 1956)

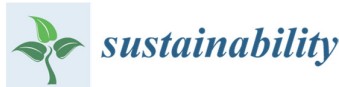
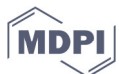

**Table 2.** Land use categories (and cropland sub-categories): area in km²; Shannon Index; average loss in Ecological Connectivity Index; geographical distribution index of land uses across counties. Source: Authors' own elaboration.

| County | Year | Urban | Cropland | Green crops | Wood crops | Vineyards | Meadows | Shrubland | Woodland | Total | Shannon (6 uses) | Average loss in ECI |
|---|---|---|---|---|---|---|---|---|---|---|---|---|
| | | | | | | Land uses, km2 | | | | | | |
| Alt Penedés | 1956 | 11.6 | 336.2 | 148.0 | 25.2 | 163.0 | 6.8 | 123.3 | 110.8 | 592.7 | 0.85 | −20% |
| | 2009 | 50.0 | 249.6 | 41.9 | 17.1 | 190.6 | 17.4 | 77.7 | 190.2 | 592.6 | 0.78 | |
| Baix llobregat | 1956 | 21.2 | 204.8 | 101.5 | 76. 0 | 27.4 | 14.0 | 104.0 | 128.2 | 485.8 | 0.87 | −42% |
| | 2009 | 146.6 | 60.2 | 34.2 | 21.9 | 4.1 | 20.3 | 87.4 | 148.0 | 485.8 | 0.65 | |
| Barcelonés | 1956 | 57.7 | 36.7 | 34.9 | 1.1 | 0.7 | 8.2 | 15.4 | 15.2 | 142.1 | 0.59 | −37% |
| | 2009 | 112.1 | 0.8 | 0.7 | 0.1 | 0.1 | 1.1 | 13.8 | 15.7 | 145.5 | 0.30 | |
| Garraf | 1956 | 5.9 | 65.8 | 27.2 | 19.3 | 19.4 | 2.5 | 70.2 | 37.1 | 184.1 | 0.84 | −31% |
| | 2009 | 39.1 | 23.6 | 9.6 | 3.5 | 10.5 | 4.7 | 56.7 | 55.0 | 184.9 | 0.67 | |
| Maresme | 1956 | 14.9 | 145.8 | 110.2 | 15.4 | 20.3 | 11.8 | 41.2 | 178.9 | 397.1 | 0.74 | −47% |
| | 2009 | 92.4 | 44.1 | 37.2 | 4.0 | 2.9 | 12.9 | 33.5 | 207.2 | 397.9 | 0.54 | |
| V. Occidental | 1956 | 29.2 | 208.3 | 154.2 | 22.4 | 31.7 | 11.7 | 65.9 | 256.1 | 582.7 | 0.74 | −42% |
| | 2009 | 165.4 | 64.6 | 56.6 | 7.5 | 0.5 | 18.3 | 67.8 | 248.7 | 583.0 | 0.56 | |
| V. Oriental | 1956 | 20.4 | 268.3 | 232.5 | 26.1 | 9.7 | 9.9 | 71.9 | 475.0 | 851.0 | 0.61 | −33% |
| | 2009 | 139.9 | 124.5 | 114.3 | 8.2 | 2.0 | 23.2 | 60.1 | 492.2 | 850.9 | 0.52 | |
| BMR | 1956 | 160.8 | 1266.0 | 808.4 | 185.5 | 272.1 | 65.0 | 491.9 | 1201.2 | 3236.0 | 0.81 | −36% |
| | 2009 | 745.6 | 567.4 | 294.5 | 62.3 | 210.7 | 98.0 | 397.0 | 1357.0 | 3240.0 | 0.67 | |
| Geographical distribution Index | 1956 | 0.89 | 0.91 | 0.91 | 0.84 | 0.67 | 0.96 | 0.94 | 0.82 | 0.93 | | |
| | 2009 | 0.95 | 0.78 | 0.82 | 0.82 | 0.23 | 0.90 | 0.95 | 0.86 | | | |

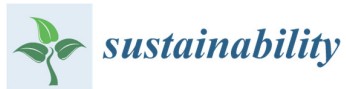
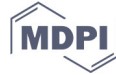

Livestock units have increased by 40%, but not in a uniform way. The most evident effect has been the near to disappearance of work animals and a general substitution of monogastric species—swines in particular—for ruminants, the latter go down from 92% of total LU500 to 39%. The other important effect is a shift in regional specialization from Barcelonès to Vallès Oriental. In 1956, the Barcelonès concentrated 32% of the RMB's LU500, mainly because of milk-producing dairy farms (because of the quick perishability of milk they had to be close to the place of consumption [37,38]). In 2009, Vallés Oriental concentrated even more LU500 (62%). As a consequence, the general livestock distribution index has decreased from 0.89 to 0.60 and that of most animal typologies has also decreased from around 0.80–0.90 to around 0.50–0.60 (Table 3).

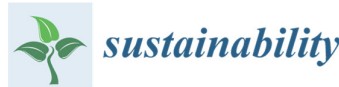
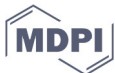

**Table 3.** Livestock composition, geographical, and typology distribution indexes. Source: Authors' own elaboration.

| County | Year | LU500 | | | | | | As a % of total LU500 | | | | |
|---|---|---|---|---|---|---|---|---|---|---|---|---|
| | | Cattle | Sheep &goats | Equids | Swines | Poultry &rabbits | Total | Cattle | Sheep &goats | Equids | Swines | Poultry &rabbits |
| Alt Penedés | 1956 | 482 | 712 | 2059 | 154 | 357 | 3765 | 13% | 19% | 55% | 4% | 9% |
| | 2009 | 2132 | 650 | 159 | 2408 | 4337 | 9686 | 22% | 7% | 2% | 25% | 45% |
| Baix llobregat | 1956 | 1591 | 442 | 2063 | 227 | 232 | 4554 | 35% | 10% | 45% | 5% | 5% |
| | 2009 | 489 | 458 | 169 | 286 | 35 | 1436 | 34% | 32% | 12% | 20% | 2% |
| Barcelonés | 1956 | 7252 | 737 | 6525 | 963 | 120 | 15,597 | 46% | 5% | 42% | 6% | 1% |
| | 2009 | 0 | 3 | 4 | 0 | 0 | 7 | 0% | 43% | 55% | 0% | 1% |
| Garraf | 1956 | 303 | 269 | 455 | 50 | 65 | 1141 | 27% | 24% | 40% | 4% | 6% |
| | 2009 | 120 | 226 | 113 | 2 | 42 | 503 | 24% | 45% | 22% | 0% | 8% |
| Maresme | 1956 | 4029 | 197 | 1971 | 141 | 235 | 6573 | 61% | 3% | 30% | 2% | 4% |
| | 2009 | 2039 | 321 | 220 | 2737 | 1077 | 6394 | 32% | 5% | 3% | 43% | 17% |
| V. Occidental | 1956 | 2645 | 416 | 2107 | 210 | 282 | 5659 | 47% | 7% | 37% | 4% | 5% |
| | 2009 | 2009 | 608 | 390 | 4225 | 594 | 7825 | 26% | 8% | 5% | 54% | 8% |
| V. Oriental | 1956 | 7087 | 531 | 2847 | 340 | 392 | 11,196 | 63% | 5% | 25% | 3% | 3% |
| | 2009 | 14,142 | 2102 | 317 | 23,910 | 1412 | 41,883 | 34% | 5% | 1% | 57% | 3% |
| BMR | 1956 | 23,389 | 3303 | 18,027 | 2084 | 1682 | 48,486 | 48% | 7% | 37% | 4% | 3% |
| | 2009 | 20,931 | 4368 | 1370 | 33,567 | 7497 | 67,733 | 31% | 6% | 2% | 50% | 11% |
| Geographical distribution | 1956 | 0.82 | 0.96 | 0.89 | 0.82 | 0.94 | 0.89 | | | | | |
| Index | 2009 | 0.55 | 0.77 | 0.88 | 0.48 | 0.60 | 0.60 | | | | | |

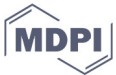

Population has also increased, but not as much as urban area, so that the allowance of urban and cropland area per capita has changed dramatically: the result is growth in the urban area per capita (+140%) and a sharp decrease in the cropland area per capita (−77%); the urban/cropland area ratio has grown from 0.13 to 1.31 (Table 4).

Farmers have decreased even more than cropland area (−88%), so that each one now cultivates on average almost 8 ha, up from just above 2 ha in 1956. Even more remarkable is the growth in LU500 per farmer: from 0.8 to 9.3, mainly because of large feedlots in the Vallès area experiencing peaks of 20 to 40 LU500/farmer. Power capacity per farmer went up from 0.6 hp—70% of which was animal power—to 90 hp in 2009 peaking at 150 to 260 in Vallès counties. Mechanization was at its beginning in 1956, machinery growth rates were in double digits and they have maintained a similar pace throughout the second half of the 20th century. In summary, endowments per farmer have increased by a factor 4 (cropland), by a factor 10 (livestock), and by a factor 150 (power capacity).

An important indicator is the evolution of livestock density measured in LU500/ha cropland, that has grown from 0.4 to 1.2, reaching 3.36 in Vallès Oriental. Instead, cropland per capita has decreased sharply. Table 4 shows the main relationships across funds for each county.

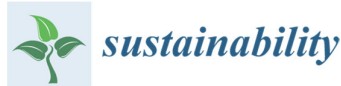
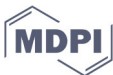

**Table 4.** Fund/fund relationships (population and farmers; livestock, area and power capacities). Source: Authors' own elaboration from the data sources given in the text.

| County | Year | Population and fund/fund relationships | | | | Farmers and fund/fund relationships | | | | | Cropland and fund/fund relationships | | | | Population (% of RMB) and distribution index |
| | | population | pop density | m2 cropland /cap | m2 urban /cap | farmers | ha cropland /farmer | LU500/ farmer | hp(m)/ farmer | hp(a)/ farmer | cropland, ha | LU500/ha | hp(m)/ha | hp(a)/ha | |
| Alt Penedés | 1956 | 47,281 | 80 | 7110 | 246 | 9095 | 3.7 | 0.4 | 0.3 | 0.3 | 33,617 | 0.11 | 0.07 | 0.09 | 0.02 |
| | 2009 | 104,353 | 176 | 2391 | 479 | 2616 | 9.5 | 3.7 | 91.0 | 0.1 | 24,956 | 0.39 | 9.54 | 0.01 | 0.02 |
| Baix llobregat | 1956 | 174,155 | 358 | 1176 | 122 | 8609 | 2.4 | 0.5 | 0.2 | 0.4 | 20,480 | 0.22 | 0.10 | 0.15 | 0.07 |
| | 2009 | 793,655 | 1634 | 76 | 185 | 918 | 6.6 | 1.6 | 64.0 | 0.3 | 6021 | 0.24 | 9.75 | 0.04 | 0.16 |
| Barcelonés | 1956 | 1,821,324 | 12,821 | 20 | 32 | 12,170 | 0.3 | 1.3 | 0.0 | 0.8 | 3674 | 4.25 | 0.10 | 2.72 | 0.71 |
| | 2009 | 2,251,600 | 15,475 | 0 | 50 | 54 | 1.5 | 0.1 | 41.9 | 0.1 | 84 | 0.08 | 27.12 | 0.07 | 0.45 |
| Garraf | 1956 | 39,869 | 217 | 1651 | 147 | 1889 | 3.5 | 0.6 | 0.2 | 0.4 | 6583 | 0.17 | 0.06 | 0.11 | 0.02 |
| | 2009 | 143,066 | 774 | 165 | 273 | 209 | 11.3 | 2.4 | 102.6 | 0.8 | 2360 | 0.21 | 9.09 | 0.07 | 0.03 |
| Maresme | 1956 | 125,660 | 316 | 1160 | 118 | 9238 | 1.6 | 0.7 | 0.1 | 0.3 | 14,582 | 0.45 | 0.09 | 0.21 | 0.05 |
| | 2009 | 426,565 | 1072 | 103 | 217 | 2044 | 2.2 | 3.1 | 38.6 | 0.2 | 4407 | 1.45 | 17.91 | 0.07 | 0.09 |
| V. Occidental | 1956 | 268,386 | 461 | 776 | 109 | 8995 | 2.3 | 0.6 | 0.2 | 0.4 | 20,832 | 0.27 | 0.08 | 0.15 | 0.10 |
| | 2009 | 878,893 | 1508 | 74 | 188 | 372 | 17.4 | 21.0 | 262.0 | 1.6 | 6462 | 1.21 | 15.08 | 0.09 | 0.18 |
| V. Oriental | 1956 | 90,058 | 106 | 2979 | 226 | 12,294 | 2.2 | 0.9 | 0.2 | 0.4 | 26,829 | 0.42 | 0.11 | 0.16 | 0.04 |
| | 2009 | 394,061 | 463 | 316 | 355 | 1095 | 11.4 | 38.3 | 148.8 | 0.4 | 12,451 | 3.36 | 13.08 | 0.04 | 0.08 |
| BMR | 1956 | 2,566,733 | 793 | 493 | 63 | 62,289 | 2.0 | 0.8 | 0.2 | 0.4 | 126,597 | 0.38 | 0.09 | 0.22 | 0.55 |
| | 2009 | 4,992,193 | 1541 | 114 | 149 | 7309 | 7.8 | 9.3 | 90.2 | 0.3 | 56,741 | 1.19 | 11.63 | 0.04 | 0.80 |

*3.2. Increased Energy Inefficiency*

Following [26,39] we present the eight EROI indicators explained in the section before (Table 5).

With the exception of Barcelonès, where Livestock has nearly disappeared, FEROI has gone down in each county from around 0.8 to around 0.2, with the highest value in Baix Llobregat (0.36). IFEROI has in general gone up (at RMB level from 0.9 to 1.3)—implying less biomass recirculation from land to cropland and to livestock. EFEROI has decreased sharply, overall from 1.2 to 0.2. In most counties, it used to be above 2 and even up to 3.6 in 1956, while in 2009 it was only between 0.14 and 0.42, meaning that the large increase in inputs (mainly machinery and imported feed) have outnumbered higher farm or livestock final productivity.

NPPEROI has gone down: while NPP has maintained, the unharvested part of NPP has increased—mainly due to forest transition, and TIC have grown exponentially in every county except Barcelonès. The Agroecological EROI has approximately halved in all counties. Biodiversity EROI has gone down slightly because, except in Barcelonès, TIC have gone dramatically up.

Land EROI has gone down from 3.27 to 0.2, with the highest value in Baix Llobregat (0.39)— where the productivity is high because 65% of cropland is irrigated versus an average of 27% in the BMR—and in Vallès Oriental (0.28) where forest extraction per hectare is highest. On the other hand, feedlots, modern feeding apt for monogastric animals and the elimination of work animals are a very linear-efficient way to fatten livestock, while in the past ruminants, open-air grazing and work animals prevailed. As a result, Livestock EROI is the only EROI indicator that has increased, from 0.05 to 0.1 (dominated by Maresme, Alt Penedès and Vallès Occidental where monogastric constitute more than 60% of LU500). From an input-output perspective livestock productivity is much lower than land productivity.

In summary, even if Land and Livestock EROI values have converged, the former is still much higher than the latter. This implies that the growth of livestock final produce over final produce (LFP/FP), up from 8% to 19%, has led to less energy efficient agriculture (see the variation in LFP/FP plotted against the variation in FEROI of Figure 3): when LFP/FP has gone down to nearly zero, as in the Barcelonès, FEROI has increased by nearly 40%; where LFP/FP has boomed, as in Alt Penedès (up from 2% to 21%), FEROI has collapsed by 80%.

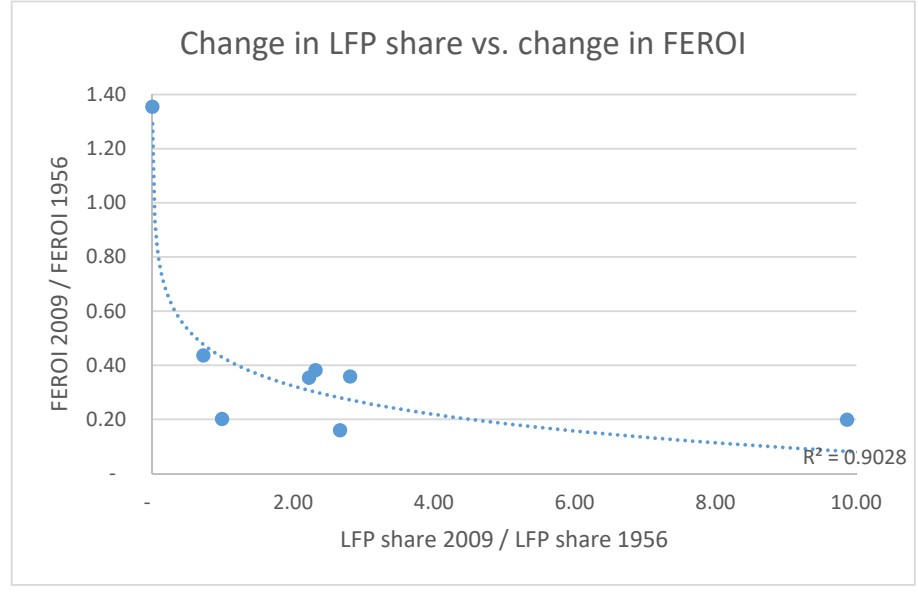

**Figure 3.** Change in livestock final produce (LFP) share vs. change in FEROI. Source: Authors' own elaboration.

*3.3. Spatially Explicit Flow-Fund Relationships: Nexus and Landscape Ecology*

Table 5 represents the overall per hectare flows of FP, BR, and EI, in which, at a glance, is possible to see how lower biomass flows are substituted for higher flows in external inputs

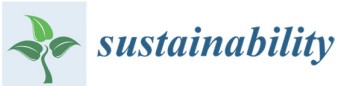 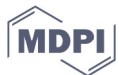

Table 5. EROI indicators, share of LFP and flow/fund relations: FP/ha, BR/ha, EI/ha. Source: Authors' own elaboration

| County | Year | FEROI | IFEROI | EFEROI | NPP EROI | AG-EROI | BiodivEROI | Land EROI | Livestock EROI | LFP share | FP/ha | BR/ha | EI/ha |
|---|---|---|---|---|---|---|---|---|---|---|---|---|---|
| Alt Penedés | 1956 | 0.88 | 1.28 | 2.81 | 1.19 | 0.30 | 0.66 | 3.30 | 0.03 | 0.02 | 13.87 | 10.80 | 4.93 |
| | 2009 | 0.17 | 2.72 | 0.19 | 0.61 | 0.12 | 0.33 | 0.16 | 0.15 | 0.21 | 17.27 | 6.35 | 92.42 |
| Baix llobregat | 1956 | 0.83 | 1.38 | 2.09 | 1.14 | 0.29 | 0.64 | 3.93 | 0.04 | 0.04 | 18.14 | 13.18 | 8.69 |
| | 2009 | 0.36 | 2.80 | 0.42 | 0.84 | 0.14 | 0.61 | 0.39 | 0.07 | 0.03 | 14.69 | 5.25 | 35.38 |
| Barcelonés | 1956 | 0.10 | 1.00 | 0.11 | 0.21 | 0.09 | 0.08 | 1.83 | 0.05 | 0.52 | 27.89 | 27.96 | 261.47 |
| | 2009 | 0.13 | 2.66 | 0.14 | 0.90 | 0.02 | 0.83 | 0.14 | 0.00 | 0.00 | 1.78 | 0.67 | 12.95 |
| Garraf | 1956 | 0.80 | 1.28 | 2.13 | 1.12 | 0.23 | 0.71 | 3.60 | 0.04 | 0.04 | 10.87 | 8.50 | 5.09 |
| | 2009 | 0.16 | 1.36 | 0.18 | 0.80 | 0.06 | 0.61 | 0.19 | 0.03 | 0.04 | 5.72 | 4.20 | 31.43 |
| Maresme | 1956 | 0.56 | 0.82 | 1.78 | 1.07 | 0.22 | 0.61 | 3.63 | 0.06 | 0.09 | 16.78 | 20.56 | 9.44 |
| | 2009 | 0.20 | 0.99 | 0.25 | 0.72 | 0.09 | 0.55 | 0.20 | 0.11 | 0.21 | 13.42 | 13.57 | 54.44 |
| V. Occidental | 1956 | 0.85 | 1.15 | 3.30 | 1.16 | 0.24 | 0.72 | 4.02 | 0.05 | 0.05 | 17.25 | 14.98 | 5.24 |
| | 2009 | 0.14 | 0.74 | 0.17 | 0.71 | 0.06 | 0.57 | 0.15 | 0.05 | 0.13 | 8.06 | 10.95 | 48.46 |
| V. Oriental | 1956 | 0.74 | 0.93 | 3.61 | 1.13 | 0.21 | 0.72 | 3.74 | 0.07 | 0.08 | 16.78 | 18.06 | 4.65 |
| | 2009 | 0.26 | 1.62 | 0.32 | 0.69 | 0.13 | 0.51 | 0.28 | 0.10 | 0.22 | 20.28 | 12.50 | 64.24 |
| BMR | 1956 | 0.51 | 0.88 | 1.20 | 1.02 | 0.20 | 0.61 | 3.27 | 0.05 | 0.08 | 14.93 | 16.91 | 12.45 |
| | 2009 | 0.19 | 1.29 | 0.23 | 0.69 | 0.10 | 0.49 | 0.20 | 0.10 | 0.19 | 13.74 | 10.63 | 60.32 |

This is a clear result of agricultural industrialization in cropping and in livestock breeding in which cheaper industrial inputs and feed substitute for local biomass flows. However, there is a more nuanced picture when these relationships are calculated for the different land uses and with respect to LU500 livestock densities (Table 6).

Productivity per hectare of major crops has increased in terms of main produce. Primary data show that the main product of cereal crops (which constituted 50% and 81% of green crop area in 1956 and 2009, respectively) has increased from 13.9 to 49.6 GJ/ha. Grapes productivity in vineyards (constituting 21% to 37% of all cropland) has also increased from 11.8 to 27.6 GJ/ha. However, in terms of biomass recirculation and final productivity, there is a lower capacity to take advantage of products from wood crops, meadows and woodland, as well as of by-products from any land use typology. For instance, in 1956, vineyards, wood crops, and woodland were also used for pasture; meadows were exploited at a higher capacity; pruned branches were used for domestic fuel needs; and, straw was eaten by working animals and used as stable beds to help manure compost.

The result is a lower per hectare productivity in certain land uses: wood crops final productivity decreased from 45 to 19 GJ/ha; woodland final productivity decreased from 17 to 12 GJ/ha, if BR is also accounted, that is if we consider the pasture in forests that was modelled only for 1956, it went down from 24 to 12 GJ/ha; an average of 41% of meadows productivity was pastured versus only 5% in 2009; wood crops and vineyard were offering also just below 1 GJ/ha for animals (by grazing or by eating leaves of pruned branches), which was not done anymore in 2009: the amount of BR for livestock has therefore been reduced in almost all land covers, but it has paradoxically increased in green crops, because of the increase in feed-oriented cropland area. Table 6 presents biomass flows per hectare of different farm use categories and per LU500.

**Table 6.** Biomass and energy flows across land uses and livestock; GJ/ha and, in parentheses, GJ/LU500. Source: Authors' own elaboration.

| County | Year | FP, GJ/ha (GJ/LU500) | | | | | | BR, GJ/ha (GJ/LU500) | | | | | | | | | EI, GJ/ha (GJ/LU500) | | | | | |
|---|---|---|---|---|---|---|---|---|---|---|---|---|---|---|---|---|---|---|---|---|---|---|
| | | Green crops | Wood crops | Vineyards | Forest | (Animal) | Total | Green crops | Wood crops | Vineyards | Meadows | Forest | From land to livestock | From livestock to cropland | (From land to livestock) | (From livestock to cropland) | Green crops | Wood crops | Vineyards | Forest | (Animal) | Total EI per farmland |
| Alt Penedés | 1956 | 17.8 | 43.1 | 13.8 | 16.9 | 4.5 | 13.9 | 36.4 | 0.6 | 0.8 | 6.9 | 4.6 | 10.2 | 13.3 | 156.4 | 118.5 | 8.2 | 9.4 | 2.5 | 1.5 | 21.7 | 4.9 |
| | 2009 | 12.9 | 16.7 | 27.6 | 6.5 | 19.8 | 17.3 | 80.4 | 0.0 | 0.0 | 1.4 | 0.0 | 5.6 | 23.6 | 30.8 | 60.9 | 168.5 | 87.4 | 150.4 | 11.2 | 104.9 | 92.5 |
| Baix llobregat | 1956 | 17.9 | 46.5 | 13.8 | 16.9 | 6.5 | 18.1 | 43.6 | 0.2 | 0.4 | 28.5 | 8.0 | 12.7 | 21.9 | 125.3 | 98.6 | 8.2 | 9.4 | 2.5 | 1.3 | 47.2 | 8.7 |
| | 2009 | 93.1 | 20.7 | 27.6 | 5.2 | 8.5 | 14.7 | 47.7 | 0.0 | 0.0 | 1.2 | 0.0 | 3.6 | 8.9 | 79.7 | 37.4 | 207.2 | 82.2 | 117.8 | 6.9 | 50.9 | 35.2 |
| Barcelonés | 1956 | 20.0 | 41.3 | 13.8 | 16.9 | 7.0 | 27.9 | 41.2 | 0.2 | 0.2 | 42.1 | 21.4 | 27.1 | 205.5 | 13.1 | 48.4 | 11.5 | 15.5 | 4.8 | 4.0 | 123.5 | 261.5 |
| | 2009 | 67.7 | 22.7 | 27.6 | 0.3 | 1.4 | 1.8 | 29.1 | 0.0 | 0.0 | 0.6 | 0.0 | 0.4 | 5.6 | 207.0 | 70.1 | 455.7 | 295.1 | 395.2 | 1.4 | 177.9 | 12.8 |
| Garraf | 1956 | 13.6 | 29.8 | 13.8 | 16.9 | 6.3 | 10.9 | 44.8 | 0.2 | 0.6 | 11.7 | 5.7 | 8.1 | 17.8 | 124.7 | 102.8 | 7.6 | 7.6 | 2.3 | 1.2 | 39.8 | 5.1 |
| | 2009 | 38.3 | 16.1 | 27.6 | 1.0 | 5.9 | 5.7 | 59.4 | 0.0 | 0.0 | 4.4 | 0.0 | 3.5 | 12.4 | 97.2 | 58.0 | 179.2 | 86.0 | 153.3 | 1.8 | 129.5 | 31.3 |
| Maresme | 1956 | 15.8 | 45.3 | 13.8 | 16.9 | 9.1 | 16.8 | 45.1 | 0.0 | 0.3 | 20.6 | 14.1 | 19.9 | 35.9 | 114.2 | 79.7 | 8.2 | 9.3 | 2.5 | 1.3 | 34.2 | 9.5 |
| | 2009 | 17.3 | 36.4 | 27.6 | 11.0 | 13.2 | 13.4 | 107.9 | 0.0 | 0.0 | 2.5 | 0.0 | 11.9 | 64.5 | 55.4 | 44.4 | 195.6 | 155.1 | 131.7 | 16.5 | 64.0 | 53.0 |
| V. Occidental | 1956 | 19.9 | 48.3 | 13.8 | 16.9 | 7.8 | 17.3 | 43.1 | 0.7 | 1.3 | 11.6 | 5.0 | 14.1 | 26.7 | 134.8 | 98.2 | 7.9 | 8.6 | 2.3 | 1.0 | 19.7 | 5.3 |
| | 2009 | 14.2 | 22.9 | 27.6 | 7.3 | 5.2 | 8.1 | 76.3 | 0.0 | 0.0 | 2.8 | 0.0 | 9.6 | 44.7 | 48.9 | 36.9 | 192.5 | 135.6 | 177.4 | 15.0 | 45.3 | 48.2 |
| V. Oriental | 1956 | 14.1 | 52.1 | 13.8 | 16.9 | 9.5 | 16.8 | 53.6 | 0.7 | 1.6 | 2.3 | 5.0 | 17.0 | 37.2 | 125.3 | 89.2 | 7.6 | 9.2 | 2.3 | 1.1 | 11.5 | 4.7 |
| | 2009 | 12.2 | 11.9 | 27.6 | 19.5 | 7.3 | 20.3 | 76.0 | 0.0 | 0.0 | 2.7 | 0.0 | 11.0 | 122.1 | 18.3 | 36.3 | 102.4 | 43.7 | 95.5 | 20.8 | 52.5 | 63.6 |
| BMR | 1956 | 11.2 | 45.2 | 13.8 | 16.9 | 7.7 | 14.9 | 50.9 | 0.4 | 0.8 | 18.8 | 6.9 | 16.1 | 30.4 | 100.6 | 79.3 | 7.7 | 9.5 | 2.6 | 1.4 | 56.4 | 12.5 |
| | 2009 | 13.9 | 19.4 | 27.6 | 11.6 | 9.4 | 13.7 | 86.6 | 0.0 | 0.0 | 2.2 | 0.0 | 9.3 | 48.7 | 33.3 | 40.8 | 151.0 | 79.6 | 126.6 | 17.4 | 60.8 | 58.2 |

Growth in green crops productivity has often resulted in slight increases in FP and large increases in BR. This is because of an increasing amount of green crops dedicated to feed (up from 31% to 72%; or from 19% to 56% with respect to total cropland area). FP/ha in green crops has increased the most in Baix Llobregat, Barcelonès, and Garraf, which are the counties where livestock population has decreased the most. Even if an increasing part of green crops is destined as BR to livestock, the near to abandonment of animal grazing and pasturing in wood crops, vineyards, meadows, and forests results in an overall loss of BR from land to livestock. In turn, this implies more feed imports from beyond the BMR boundaries [40].

## 4. Discussion

### 4.1. Methodological Contribtion: How Bridging MuSIASEM with ELIA Contributes to Landscape Agroecology

Our research is part of an energy–landscape integrated approach (ELIA) [10,41], in which we explicit biomass and energy flows across different land uses (MuSIASEM). The combination of ELIA and MuSIASEM provides solid foundations for a better understanding of landscape agroecology that can inform policies for the design of agro-ecological landscapes. To the notion of multi-functionality—one fund (i.e., land use) provides multiple flows that can serve multiple purposes (i.e., food, fuel, feed), we integrate that of inter-functionality: agro-ecological practices require interdependent relationships between farmers, livestock, land-uses, and productive capacity, which the application of flow-fund analysis can assess at landscape scale. By combining these methodological frameworks, we are able to represent agro-ecosystems dynamics at the adequate functional unit of analysis, and hence, offer a more articulated perspective in the new landscape agroecology paradigm. Landscape Agroecology is not a new term [4,42], it focusses on multifunctional relationships in landscapes [43] and also on interdisciplinary, multi-scale analysis [44]. It is linked to the notion of agrarian multi-functionality which, differently from Vranken et al.'s [45] understanding of agricultural multi-functionality as a provider of outputs (ecosystem services to society), it relates to the metabolic perspective on an agroecosystem (level n of analysis) and the functions performed by its internal components (level n − 1 and n − 2 of analysis) whose flows allow for both the maintenance of an agroecosystem's fund elements (internal loops) and provide services to society (final flows).

For several reasons, ELIA-based landscape agroecology is a novel application of MuSIASEM [9,33]: for its flow-fund analytical approach, for the inter-functionality between funds, and for the integration of landscape functions and services, agrarian studies, and social metabolism. Moreover, MuSIASEM applies nexus analysis [46], and here we do so by exploring the multiple nexuses between livestock, land-uses, and landscapes. In summary, the multi-scale spatial explicitation of energy flows and the nexus across different agroecosystem funds are the fundamental bricks of landscape agroecology.

### 4.2. Critical Discussion: How Agricultural Inter-Functionality and Landscape Functional Structure Have Been Disintegrated by Industrialization, Urbanization and Geographical Specialization

Industrialization of farming activities is visible in the fall of EROIs, primarily because of the losses in multi-functionality of agrarian flows and in inter-functionality across funds. However, for the specific case of livestock breeding, industrialization has implied an improvement in the Livestock EROI: but in this case, the costs are visible on the landscape.

Feedlots' economic linear-efficiency—i.e., increased livestock EROI is possible only through grain-based animal diets, stabling, and growth in the proportion of monogastric animals (as well as by economies of scale in feedlots size). This process is in antithesis with agro-ecological and agricultural landscape efficiency, even more so when livestock densities to cropland have changed this delicate equilibrium, and when machinery substitutes for work animals, which in turn explains the lower Land EROI. As a consequence of substituting feedlots for open-air grazing, and of

substituting fossil fuel burning for wooden biomass, the aggregate result, from a circular bioeconomy standpoint, is lower agroecosystem and agricultural landscape functionality. In particular, a seemingly win-win situation that is constituted by a higher availability of animal products and fuel for a cheaper market price (economic efficiency) is only the tip of an iceberg whose underneath hides a lose-lose-lose-lose-lose-lose-lose (lose[7]) socio-cultural and environmental reality with negative impacts:

1. As pastures in woods and in meadows are abandoned, animal breeding relies more on crop-based consumption of human-edible biomass, therefore posing a threat to food sovereignty.
2. As forests are abandoned also as a source of fuel, the risk of wildfires is increased.
3. As meadows are abandoned and afforestation processes initiate, landscape heterogeneity decreases, therefore creating a loss of habitat differentiation and of bio-cultural heritage.
4. As work animals are replaced by machinery, more energy inputs are required for farmland labour; in turn, less biomass is reused. Its reuses were important from a landscape perspective as they could integrate funds one another in complex landscape mosaics.
5. As less ruminants in livestock composition, further competition for cropland main produce is exerted, because straw as a by-product can hardly be digested. (Modern cereal varieties tend to be short-stemmed in order to maximize the grain/straw ratio so that, in principle, there is less need for straw-digesting ruminants. However, these varieties show lower Net Primary Productivity –hence less carbon sequestration potential-[47], can be less nutritional than traditional long-stemmed varieties, and reduce associated biodiversity such as certain bird typologies which find shelter in tall straw cereal crops. As a consequence, these traditional varieties that in terms of grain produced are economically less efficient actually perform better with respect to carbon sequestration, water efficiency, nutritional values, associated biodiversity and potentially contribute to higher farm animal diversity and lower competition for cropland main produce).
6. As livestock density increases—most of them live now in densely populated feedlots, so that we can name this process "urbanization of livestock"—management of slurry implies groundwater pollution. The Nitrogen balance of our analysis shows for 28 municipalities an excess of 170 kgN/ha of cropland, particularly in the Vallès and Maresme counties—a widespread case in Catalonia [48].
7. As more meat in our diet, health problems and hazards increase, from high cholesterol to cancer [5], which means that the consumption of red meat should be reduced in Western countries by 78% (that is by 113 g/day) in order to meet the recommendations from WHO [49].

By considering multi-scalarity at the geographical level we can observe a process of county specialization in certain crops or functions that makes it even harder to resolve the before mentioned dysfunctionalities in the land-livestock-cropland nexus within the BMR. The Shannon-Wiener index shows how landscapes in the seven counties have become less heterogeneous. Moreover, the modified application of the index (to show how heterogeneously funds are distributed) indicates that agroecosystems of the BMR are also less functional. Tables 2 and 3 show that both cropland and livestock were less evenly distributed in 2009 than in 1956—with Alt Penedès concentrating most cropland and Vallès Oriental most livestock—while urban land, woodland, and population (Table 4) were more evenly distributed.

The growth in urban area [31] is common to Mediterranean cities, and, because of the important role played by county capitals, it assumes the form of polycentric urbanism [50], in which Barcelonès' share of urban area has gone down from 36% to 15%. However, it has come at the cost of agricultural abandonment in all counties, implying that widespread afforestation and agricultural specialization: forest transition has occurred at all scales, from four municipalities within the BMR [11,51] to Catalonia overall [52].

Vineyards in Alt Penedès (that is a specific agricultural category for a specific county) constitute 34% of BMR total cropland area, which is the only case of cropland growth. Similarly, livestock has moved from large urban centres to the periphery, and pigs in Vallès Oriental constitute 35% of BMR

total LU500. Moreover, its population of nearly 100,000 pigs was concentrated in only 111 factories, when, in 1999, they were 262. Two phenomena of productive specialization and territorial concentration that resemble more industrial districts than agroecological landscapes. From the landscape perspective, Table 2 shows that its functional structure (understood as a heterogeneous and well connected land-matrix) has also decreased and so too has the Shannon Index and the Ecological Connectivity Index [18].

### 4.3. Policy Perspective: From Organic Agriculture to Landscape Agroecology

The analysis just presented widens the spectrum for the meaning and understanding of landscape agroecology: not only organic agriculture is necessary, but, since agricultural activities are imprinted on the landscape and the landscape offers functions and services to the agriculture, it is important to analyse agroecosystems in their adequate unit of analysis—to close the metabolic cycles. Advocating for organic meat or vineyards is important, but even more so is to look at the landscape effects of widespread monocultures, and at the agro-ecological opportunities of having inter-functional vineyards and pasturing livestock. The size of the land and the livestock sub-systems needs to be well-balanced, not only in the dimension of the incoming and outgoing flows, but also in the diversity of land uses and animal typologies that are involved. In this way, dependence on External Inputs is minimized, the agroecosystem is not dysfunctional, and it resembles more a sustainable organism—closing its metabolic cycles [53].

It is important to design land-use, environmental and agricultural policies that consider how agroecosystems are inter-functional and require balance between their funds.

Moreover, a balance between flows—in particular, Final Product is also required: in agro-ecological landscapes, it will be unsustainable to have an excessive share of animal produce in relation to vegetal produce; to this extent, the change towards meat-based diets should be drastically reversed, as local agroecosystem cannot supply enough feed neither they have capacity for safely assimilating all livestock waste (Table 5.1 in [40] show changes in Catalan diet between 1956 and 1999 in which caloric and fresh weight consumption of meat, eggs, milk and cheese have both gone up by more than 90%. In 2009, animal products constituted 24% in weight and 35% in expenditure of the average Catalan household food budget [54]. Ref. [51] relates the arrival of gas bottles for cooking with abandonment of forest extractions and the increased dependence on fossil fuels, and in the same vein [36] shows how wood-crops have lost multi-functionality primarily because pruned branches are not used as FP today). Proposals such as Meatless Monday (https://www.meatlessmonday.com/) that aim at reversing dietary change should be supported by strong policies.

In traditional agriculture livestock husbandry was complementary to farming land, but now the organization of entire agroecosystems is only centred on increasing livestock. Land EROI has decreased mainly because of mechanization, which can be seen as an efficient way of animal husbandry focused solely on fattening animals for protein intake ready to be marketed, rather than on work animals whose role was multifunctional (draught power and manure, wool, leather, horn, as well as food) and helped a lot to close the cycles of agroecosystem's reproductivity. Pastures in meadows and forests are abandoned because moving animals to and from them would make them burn too many calories, slow their pace of growth diminishing this linear single-minded productivity, and raise the prices of animal products. Soil fertility is maintained with the application of energy inefficient synthetic fertilizers, while at the same time, but in other distant parts of the agroecosystem, the water table is polluted because of excessive Nitrogen lixiviated from slurry: with regional specialization it has become impossible to move these diluted nutrients to the soils where are most needed, i.e., from Vallès Oriental to the irrigated horticulture of Baix Llobregat. Therefore, we call for policies that penalize feedlots practices, and, on the other hand, give incentive to pastoral activities that are beneficial for landscapes.

Thus, in an agro-ecological model, such as it used to be in 1956, livestock was at the service of the land(scape): it was moving nutrients from pastures to cropland, maintaining landscape mosaics, and employed in cropland activities. In the 2009 case, livestock fattening has been disintegrated from agriculture and it concentrated into a linear industrial process: the land(scape) is at the service of

livestock. Tables 5 and 6 above and the growth in feed-oriented crops show how the productivity gains of agrarian industrialization have been absorbed mainly by livestock feeding, which is using up most of green crops produce (from 48 to 76 GJ/ha), while increases in food-oriented FP have been minimal (from 11 to 14 GJ/ha). Considering cropland area loss and population increase, we can claim that the provisioning ecosystem services from land have dramatically decreased as a result. At the same time, the Mediterranean landscape mosaic needs to be preserved as a basic fund in order to maintain support, regulating, and cultural ecosystem services.

In summary, it is necessary to resolve the trade-off between economic viability and land(scape) requirements by acknowledging that animal husbandry plays a key role in this dilemma. Nor can be they too many, as in the 2009 case, neither agro-ecosystems can do without them.

### 4.4. Remarks for Further Research

Given the boundary limits of our analysis, we are not covering the domestic need for cooking, neither for space heating. However, would that issue be considered on this study, then energy inefficiencies would have been even greater as a result of further dysfunctionalities in the agroecosystem and population nexuses.

A proper assessment of the energy efficiency of work animals as compared to small and large machinery goes beyond the scope of our paper, and it is a complex issue. Work animals are ruminants, so that some metabolic benefits of re-introducing them are connected to the point previously made; furthermore, in traditional rural housing systems they have often been placed underneath the sleeping rooms, so that they were a source of heat in the winter; have a lower power capacity than machinery, both for working land and as transportation means; and finally, they reproduce themselves and do not require an industrial system for their re-production. Conversely, machinery can be turned off when not needed, and, from the economic perspective, is an important labour saving technique: once more, the energy vs. economic efficiency trade-off has favoured the adoption of the less multi-functional solution.

Another not yet explored issue, and in line with Scheidel and Gerber's proposal of bridging analysis of social metabolism with needs theory [34], is the one relating agrarian social metabolism with the amount of needs satisfied in different societal contexts: a fully organic system was not only more energy efficient per se, but in that context of a frugal society, it allowed for the full satisfaction of dietary, transportation, and domestic heating needs. Instead, the agro-industrial regime within a consumerist society implies that less societal needs are satisfied from nearby agroecosystems, and therefore the metabolic stress is shifted to other places or to future generations. Notably, in the food vs. fuel dilemma the case of agro-fuels relates the societal need of transportation with political ecologies of land grabbing [55] and a metabolic shift to distant places; the case of domestic space heating—satisfied with the use of fossil fuels instead than local biomass—exemplifies a metabolic shift to future generations.

## 5. Conclusions

This paper has analysed the unsustainable path that was undertaken by agriculture in the BMR in the process of agricultural industrialization. The economic benefits of specialization in terms of increased labour and land productivity, particularly in the livestock sector, imply significant environmental costs in terms of energy inefficiency, loss of multi-functionality, and reduced product diversification. From a circular bio-economy perspective, restoring abandoned cropland, shifting livestock from feedlots back to pastures, and rescaling livestock densities are first-order priorities. From a sustainability perspective, change in diets is also very much required, particularly for the incompatible trade-off with population growth.

The application of the Multi-EROI method within MuSIASEM, and with spatial land metrics is a powerful methodological approach. In particular, our application of the Energy–Landscape Integrated Analysis sheds new lights on landscape agroecology. We have shown how the industrial model has broken the nexuses between agroecosystem funds, something that constitutes the hard

core of serious sustainability problems. Not only a more energy efficient and wildlife-friendly agriculture is required, such as organic agriculture, but this needs to be sustainable from the landscape agroecological perspective in order to better close the agroecosystem metabolic cycles and to preserve the ecological functions and services for more sustainable farm systems. This requires that internal flows are highly integrated again between those funds across different land-covers, livestock, land uses, and population work in synergy with each other.

If landscape functional structure is to be well kept sustainable, farm systems require an agro-ecological approach in a way that goes beyond organic agricultural practices considered in isolation from their territorial effects. This becomes the main task for agri-food sustainability: a step forward from organic farms to agro-ecological territories where the main biophysical cycles will begin to close [56].

**Supplementary Materials:** The following are available online at www.mdpi.com/xxx/s1, supplementary information detailing the method followed.

**Author Contributions:** C.C. has performed research and written the main part of the text; J.M. and E.T. have participated in conceiving the approach and writing the text.

**Funding:** We thank funding from the projects "Sustainable Farm Systems: Long-Term Socio-Ecological Metabolism in Western Agriculture" (Canada's Social Sciences and Humanities Research Council Partnership Grant 895-2011-1020), HAR2016-76814-C2-1-P (AEI/FEDER UE) of the Ministerio de Economía y Competitividad, Spain and "Sustainable Farm Systems and Transitions in Agricultural Metabolism" (Spain's Science Ministry grant HAR2012-38920-C02-02) which has also funded the cost of this publication in Open Access.

**Acknowledgments:** We thank Francesc Coll for providing maps and three anonymous reviewers.

**Conflicts of Interest:** The authors declare no conflict of interest.

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
