# Peer review of "Landscape Agroecology. The Dysfunctionalities of Industrial Agriculture and the Loss of the Circular Bioeconomy in the Barcelona Region, 1956–2009"

_sustainability, doi:10.3390/su10124722_

Reviewer 1 Report

Dear Authors,

In my opinion, the research is well conceived and may contribute to the discussion towards agricultural sustainability.

My comments are as follows:

-        The article reads easily, though sentence simplification (mostly shortening) may help

-        In case your research is the source for figures/tables, you may say “Authors’ own elaboration”

-        The article is very long. The methods section and the discussion section may benefit from tables or figures (e.g. stepwise research design; system boundaries; interdependences among research findings and possible role of policy makers) for supporting text synthesis. Further explanations may be appended to the main manuscript text.

-        Introduction: please aim at objectivity while avoiding any possible ideological interpretation; please go straight to the point and present the problem your research is dealing with; definitions of key concepts are required to allow non-specialists to understand;

-        In my opinion, a literature review paragraph is needed to help the reader to understand what research gap is your article aiming at bridging (this is not clear to me);

-        Sustainability assessment would imply considering the three dimensions of sustainability; please clearly present the indicators per dimension;

-        To what extent is your research useful for policy makers? What policy measures would you propose, which may base on your research findings?

Author Response

The article reads easily, though sentence simplification (mostly shortening) may help done

-        In case your research is the source for figures/tables, you may say “Authors’ own elaboration” done

-        The article is very long. The methods section and the discussion section may benefit from tables or figures (e.g. stepwise research design; system boundaries; interdependences among research findings and possible role of policy makers) for supporting text synthesis. Further explanations may be appended to the main manuscript text.

We have shortened the methods section (2.2) by creating a supplementary information document. In turn, in the manuscript we have explained in a less technical language the terminology and method of social metabolism and have added a new table with the data sources. As well, we have added in the same section a footnote specifying system boundaries.

The discussion section has been radically restructured, please see our reply to the last comment below.

-        Introduction: please aim at objectivity while avoiding any possible ideological interpretation; please go straight to the point and present the problem your research is dealing with; definitions of key concepts are required to allow non-specialists to understand; 

We have deleted references to any ideological bias that we were able to identify, if some are still visibile, please specify which ones they are. We have tried to be as objective as possible by sustaining statements with references. If there are still statements that are unreferenced and that might lead to possible interpretations as non-objective, please specify which ones they are.

We have structured the introduction following the journal instructions (https://www.mdpi.com/journal/sustainability/instructions),in which the problem we present are the sustainability costs of industrial agriculture and dietary changes, namely: "hidden costs in terms energy efficiency, landscape ecology, bio-cultural heritage, biodiversity, climate change, soil and water quality, and human nutrition and health ".  

we have deleted from the introduction mentions to social metabolism, flow/fund, fund elements which might not be understood by the non-specialist. References to their definition are now in the methodology section (2.2 Methods), see the response provided to the point above.

-        In my opinion, a literature review paragraph is needed to help the reader to understand what research gap is your article aiming at bridging (this is not clear to me);

We consider that the literature review paragrpah was already there, to this we have added one explaining the research gap. Now the old and new paragraphs are:

The current state of research in landscape agroecology is set in the outcomes of the Sustainable Farm Systems research project, which has focused on an innovative development of EROI analysis [29, 41, 69, 71] also merging it with MuSIASEM [74], on nutrient cycles [72, 73] and on the original development of Energy-Landscape Integrated Analysis [40-43], which in turn draw from previous work on social metabolism [20, 75, 77] and EROI analysis [76]. As well, these novel methods applied in landscape agroecology can bridge with and contribute to the land sharing/sparing debate [78] which is still a controversial issue [79, 80].

In particular, we learn from Tello et al., [57] that biomass flows within agro-ecosystem have to be balanced between incoming-outgoing and recirculating ones, and that industrial agriculture has favoured maximizing linear flows (input-output) over the ones that recirculate internally. But little is said about the relationships between these flows and the different funds within an agroecosystem. In this paper we fill this research gap by applying the flow/fund analysis of MuSIASEM and show that the balance between the different flows that enter into, recirculate across and exit from the different funds (i.e livestock and land-uses) is also important.”

If further literature review is required, please specify.

-        Sustainability assessment would imply considering the three dimensions of sustainability; please clearly present the indicators per dimension;

although it is common practice to reduce sustainability to 3 dimensions (economic, social and environmental), this might not always be the case. i.e. the ethical or moral dimension could be also added, or two instead of three could be considered.  

For our case, we look at social and environmental sustainability, assuming that the present industrial system of agriculture is looking at financial viability only. The result, evident from the paper, is both an environmental problem (energy inefficiency and loss of habitats) as well as a social one (loss of cultural landscapes). We have considered the issue of expliciting the sustainability dimensions and, where possible, some mention to them has been made in the text.

-        To what extent is your research useful for policy makers? What policy measures would you propose, which may base on your research findings?

The discussion section has been radically re-structured, by giving a re-orientation towards policy relevance (section 4.3) by clarifying the methodological contribution of the paper (section 4.1) and the critical analysis of the results (4.2). A new section has been added (4.4) that focusses on further research. It has been cut and, where not possible, information has been moved to footnotes.

we thank you for your comments and consider that the manuscript is much better now

Reviewer 2 Report

The designation Green Revolution to indicate the process of development of agriculture, applied to the one observed in Barcelona, does not seem happy to me (lines 19, 31…), since the reference to the Green Revolution refers to the process occurred in developing countries, particularly in Asia. I suppose the indication referred to in the conclusion (line 568)  “ agricultural industrialization” is more appropriate.

Line (227) where is Fig.1 should be Fig.2

Throughout the text and especially in the presentation of the results, several concepts of the methodology used are systematically referred to the bibliography, sometimes without a complete presentation / explanation in the article, which makes it very difficult for us to read, and above all compromises understanding of the reading and interpretation of results.

I point out here some examples of the lines in which I think should be included more information and details, not referring exclusively to the bibliography:

(lines 274, 275, 276) Shanon Index of non-sealed land uses; average loss in Ecological Connectivity Index; Geographical distribution

(line 308 to line 336) eight EROI indicators

Author Response

The designation Green Revolution to indicate the process of development of agriculture, applied to the one observed in Barcelona, does not seem happy to me (lines 19, 31…), since the reference to the Green Revolution refers to the process occurred in developing countries, particularly in Asia. I suppose the indication referred to in the conclusion (line 568)  “ agricultural industrialization” is more appropriate.

Reference to green revolution has been changed to agrarian industrialization.

Line (227) where is Fig.1 should be Fig.2 done

Throughout the text and especially in the presentation of the results, several concepts of the methodology used are systematically referred to the bibliography, sometimes without a complete presentation / explanation in the article, which makes it very difficult for us to read, and above all compromises understanding of the reading and interpretation of results.

this comment is in line with that of another reviewer so that the method section has been simplified dramatically and instead a new file added as supplementary material, we hope now the text is much more fluid, simple and comprehensible than before. 

so that we have added the following explanation: the EROI presented are i) the Final EROI, relating the Final Produce (FP) and the Total Inputs Consumed (TIC), which are the sum of Biomass Reused (BR) and External Inputs (EI); ii) the External Final EROI, relating FP/EI –which represents the relationship between the output and the input to the farm; iii) the Internal Final EROI, relating FP/BR which indicates the biomass recycling effort. Then, we analyze iv) the NPPEROI, which relates Net Primary Productivity with the sum of total inputs consumed (TIC) and unharvested biomass; v) the Agroecological EROI, which relates FP to the sum of unharvested biomass and TIC; vi) Biodiversity EROI, which relates unharvested biomass to the sum of TIC and unharvested biomass. Finally, we consider vii) Land Final EROI and viii) Livestock Final EROI representing the relationship between the Final Produce of, respectively, the Land and Livestock subsystems with the correspondent internal and external inputs. Since all these indicators relate different flows with each other, but in social metabolism it is important to relate these flows to their correspondent funds (i.e. the flow of final produce per hectare, or per farmer, or per unit of livestock) as well as the relations across funds (i.e. the size of cropland with respect to total land, or the density of livestock oer unit of cropland). By all these flows and funds relationships we are exploring the nexus between farmland and livestock functions.

I point out here some examples of the lines in which I think should be included more information and details, not referring exclusively to the bibliography:

(lines 274, 275, 276) Shanon Index of non-sealed land uses; average loss in Ecological Connectivity IndexGeographical distribution we have added more text to explain these indicators

(line 308 to line 336) eight EROI indicators each one is now explained in the methods section, please see above

Reviewer 3 Report

The submission is good in conceptualization. Subject addressed in this article is worthy of investigation. The used methodology is accurate. The article was prepared according to the requirements. The author used specific research methods and this has been preceded by a discussion of their selection. An extensive review of the literature related to the analyzed subject was presented.

Author Response

the introduction has been modified, following the reviewers' suggestions

Round  2

Reviewer 1 Report

Dear Authors,

I think the article is suitable for publication. I just suggest to check for typos and to slightly revise the language to get rid of "absolute" words, such as e.g. the verb to demonstrate.